# Variable Clinical Appearance of the Kir2.1 Rare Variants in Russian Patients with Long QT Syndrome

**DOI:** 10.3390/genes13040559

**Published:** 2022-03-22

**Authors:** Elena Zaklyazminskaya, Margarita Polyak, Anna Shestak, Mariam Sadekova, Vera Komoliatova, Irina Kiseleva, Leonid Makarov, Dmitriy Podolyak, Grigory Glukhov, Han Zhang, Denis Abramochkin, Olga S. Sokolova

**Affiliations:** 1Medical Genetics Laboratory, B.V. Petrovsky National Research Center of Surgery, 119991 Moscow, Russia; zaklyazminskaya.ev@med.ru (E.Z.); margaritapolyak@gmail.com (M.P.); anna.shestak87@gmail.com (A.S.); marisadekova@gmail.com (M.S.); dimap-cardio@mail.ru (D.P.); 2Center for Syncope and Cardiac Arrhythmias in Children and Adolescents, Federal Medical Biological Agency, 115481 Moscow, Russia; verakom@list.ru (V.K.); vkis2@yandex.ru (I.K.); dr.leonidmakarov@mail.ru (L.M.); 3Faculty of Biology, Shenzhen MSU-BIT University, Shenzhen 517182, China; gluhovg@gmail.com (G.G.); zhanghan@smbu.edu.cn (H.Z.); 4Faculty of Biology, Lomonosov Moscow State University, 119234 Moscow, Russia; abram340@mail.ru; 5Laboratory of Cardiac Electrophysiology, National Medical Research Center for Cardiology, 121500 Moscow, Russia

**Keywords:** primary arrhythmia, Long QT syndrome, LQT7, Andersen-Tawil syndrome, Kir2.1, *KCNJ2*, atrial fibrillation, variant reassessment

## Abstract

Background: The *KCNJ2* gene encodes inward rectifier Kir2.1 channels, maintaining resting potential and cell excitability. Presumably, clinical phenotypes of mutation carriers correlate with ion permeability defects. Loss-of-function mutations lead to QTc prolongation with variable dysmorphic features, whereas gain-of-function mutations cause short QT syndrome and/or atrial fibrillation. Methods: We screened 210 probands with Long QT syndrome for mutations in the *KCNJ2* gene. The electrophysiological study was performed for the p.Val93Ile variant in the transfected CHO-K1 cells. Results: We found three rare genetic variants, p.Arg67Trp, p.Val93Ile, and p.R218Q, in three unrelated LQTS probands. Probands with p.Arg67Trp and p.R218Q had a phenotype typical for Andersen-Tawil (ATS), and the p.Val93Ile carrier had lone QTc prolongation. Variant p.Val93Ile was initially described as a gain-of-function pathogenic mutation causing familial atrial fibrillation. We validated electrophysiological features of this variant in CHO-K1 cells, but no family members of these patients had atrial fibrillation. Using ACMG (2015) criteria, we re-assessed this variant as a variant of unknown significance (class III). Conclusions: LQT7 is a rare form of LQTS in Russia, and accounts for 1% of the LQTS cohort. Variant p.Val93Ile leads to a gain-of-function effect in the different cell lines, but its clinical appearance is not so consistent. The clinical significance of this variant might be overestimated.

## 1. Introduction

The electrophysiological characteristics of Kir channels determine their role in maintaining resting potential and regulating cell excitability. Under physiological conditions, Kir channels pass high-amplitude potassium currents when the membrane is hyperpolarized, and are inactive when depolarized. These channels can be compared to diodes, conducting current in one direction. When the expression level of potassium channels of internal rectification is high, the resting potential of cells will be close to potassium equilibrium potential [1], and no spontaneous electrical activity will be observed in them.

The Kir family consists of seven types of channels. Mutations in the genes encoding Kir1–Kir7 lead to multisystem pathologies: ventricular arrhythmias, retinal diseases, intermittent paralysis, and diabetes mellitus [2,3,4,5]. The *KCNJ2* gene (17q24.3) encodes the inward rectifier Kir2.1 potassium channel [3], which maintains resting potential and cell excitability in cardiomyocytes, nerves, and muscles, and plays an important role in craniofacial morphogenesis [6].

The channel pore is formed as a tetramer of subunits [5]. Each subunit has two helical transmembrane domains, M1 (aminoacids 82–106) and M2 (157–178), which are separated by a pore-loop with extracellular (107–128), intramembrane pore-forming (129–147), extracellular (148–156) segments, and two intracellular domains linked to the M1 (N-terminus, 1–81) and M2 (C-terminus, 179–427) [5,7].

The first study reporting dominant-negative mutations in the *KCNJ2* gene in LQTS/ATS was published in 2001 [8]. Genetic alterations in the *KCNJ2* gene affecting the Kir 2.1 protein may lead to several pro-arrhythmic disorders with an autosomal dominant mode of inheritance [3]. Presumably, the clinical phenotype in patients with a mutated *KCNJ2* gene correlates with ion permeability defects. Loss-of-function mutations lead to QTc prolongation (LQT7), with variable dysmorphic features and periodic paralysis Andersen-Tawil syndrome [5,6]. In contrast, gain-of-function mutations cause familial atrial fibrillation (MIM#613980). Additionally, several gain-of-function variants were found in patients with Short QT syndrome (SQTS) [5,9]. According to the ClinVar database, 330 variants in total are known for the *KCNJ2* gene, and 94 variants are listed as “likely pathogenic” and “pathogenic” [10]. The whole phenotypic spectrum of the *KCNJ2* mutations is still subject to clarification [11], due to the rareness of the condition.

In this paper, we present results of the genetic screening of the *KCNJ2* gene in a Russian cohort of LQTS patients, and discuss the complexity of interpretation of genetic variants.

## 2. Materials and Methods

We performed a clinical and genetic investigation of the 210 unrelated probands with a clinical diagnosis of Long QT syndrome (LQTS). Diagnosis was established based on Schwartz’s diagnostic score [12].

Clinical and genetic investigations were performed in accordance with the Helsinki declaration and based on written informed consent for clinical and genetic testing, interventional treatment (if needed), and publication. All genetic testing procedures were performed according to the current clinical guidelines and by direct request from family members, with written informed consent of the full-aged family members, or of their legal representatives if under 18 years. All data presented in the manuscript were properly anonymized. The manuscript was presented at the Local Ethic Committee on 2 February 2022, and considered appropriate for publication (Protocol No. 04-2022 on 2 February 2022). The clinical research included general examination, family, and personal history-taking, blood electrolytes measurements, repeated resting and standing ECG, 24 h Holter Monitoring of ECG, and EcoCG.

Genomic DNA was extracted from the venous blood. Extraction and purification were performed using a Quick-DNA Miniprep Plus Kit (Zymo Research Corp., Irvine, CA, USA), according to the manufacturer’s instructions. The genetic study for all probands included target gene panel sequencing for 11 genes (*KCNQ1, KCNH2, SCN5A, KCNE1, KCNE2, KCNJ2, SNTA1, SCN1B-4B* genes) based on the IonTorrent platform (Thermo Fisher Scientific, Waltham, MA, USA). For proband LQTS_139, additional genetic testing was performed with whole-exome sequencing (WES) using DNBSEQ-G400 (MGI, BGI Group, Shenzhen, China) with SureSelect Human All Exon V7 library preparation kits (Agilent, Santa Clara, CA, USA). Reads were aligned to the GRCh37/UCSC hg19 human genome build. Sequence variants were analyzed using a custom-developed bioinformatics pipeline. According to the standard protocol, all variants found by WES were confirmed by bi-directional capillary Sanger resequencing on an ABI 3730XL DNA Analyzer (Thermo Fisher Scientific, Waltham, MA, USA). Pathogenicity assessment for all rare genetic variants was performed according to ACMG2015 guidelines [13]. De novo status of mutation was confirmed according to ACMG2015 recommendations. Relatives of probands were invited for genetic counseling and cascade familial screening if any rare variant(s) of class III-V (VUS/likely pathogenic/pathogenic) were found. Cascade familial screening was performed by bi-directional capillary Sanger resequencing on an ABI 3730XL DNA Analyzer (Thermo Fisher Scientific, Waltham, MA, USA) using the target gene(s) fragment.

Mutation c.277G>A (p.Val93Ile) was introduced into pcDNA3.1-Kir 2.1 plasmids by PCR-based site-directed mutagenesis. Plasmid pEGFP-N1 containing green fluorescent protein (GFP) was used to visualize transfected mammalian cells (CHO-K1). CHO-K1 cells were co-transfected with two plasmids: pcDNA3.1 encoding WT or V93I-Kir2.1 channel, and pEGFP-N1 encoding GFP at a 5:1 ratio. Cells were transfected using Lipofectamine LTX and Plus reagent (Invitrogen, Thermo Fisher Scientific, Waltham, MA, USA). The CHO-K1 cells on a small coverslip were placed in an experimental chamber with a constant flow of physiological solution (mmol/L: 150 NaCl, 5.4 KCl, 1.8 CaCl2, 1.2 MgCl2, 10 glucose, 10 HEPES, pH was adjusted to 7.6 with NaOH), placed on the stage of an Eclipse Ti-S (Nikon, Japan) inverted fluorescence microscope. Currents were recorded at room temperature (24 ± 0.5 °C) with the help of an Axopatch 200B amplifier (Molecular Devices, San Jose, CA, USA). When irradiated with 480 nm excitation light, the cells emitting green fluorescent light were selected for recording. Patch pipettes of 1.5–2.5 MOhm resistance were made of borosilicate glass (Sutter Instrument, Novato, CA, USA) and filled with the solution (mmol/L): 140 KCl, 1 MgCl_2_, 5 EGTA, 4 MgATP, 0.3 Na_2_GTP, 10 HEPES; pH has been adjusted to 7.2 with KOH. Access resistance and capacitances of pipette and cell were routinely compensated before the start of each recording. As a result, current amplitudes were normalized to the capacitive cell size (pA/pF).

The results were statistically processed using a GraphPad Prism 7. In addition, the significance of differences in IK1 parameters between groups of cells at different values of membrane potential was determined by ANOVA with Tukey post hoc test.

## 3. Results

We found three LQTS probands carrying rare heterozygous genetic variants p.Arg67Trp, p.Val93Ile, and p.Arg218Gln in the *KCNJ2* gene.

The clinical and genetic characteristics of patients are summarized in Table 1.

Proband LQTS_96, 29 y.o., female, experienced cardiac arrest and had a cardioverter-defibrillator implanted for secondary prophylaxis of sudden cardiac death (SCD). During the examination, QTc prolongation was detected up to 494–500 ms at the resting ECG, and no structural abnormality was found during a cardiac ultrasound examination. The potassium level was average, 3.8–4.2 mmol/L, and the patient did not report any remarkable episodes of muscular weakness. Extra-cardiac phenotype had included short stature, low-set and rotated ears, hypertelorism, epicanthus, small mandibulae and chin, clinodactyly of the IV-V finders, small hands and feet, and mild scoliosis. β-blockers (atenolol) were prescribed, and she had no appropriate shocks during three years of follow-up. Known pathogenic genetic variant c.199C>T (p.Arg67Trp) in the *KCNJ2* gene was found. Healthy parents and son (3 y.o.) were tested, and none carried this variant.

Proband LQTS_33, female, 23 y.o., had syncopal episodes since adolescence and documented episodes of VT/VF (Figure 1). She also reported one episode of muscular weakness after a long-lasting episode of unconsciousness, but without evidence of hypokalemia. At the time of hospitalization, she had her usual potassium level at 3.2 mmol/L, which is relatively low but within the normal range. QTc duration was exceedingly prolonged up to 600–620 ms at the resting ECG and 650 ms when standing. No structural abnormality was found during the cardiac ultrasound examination. She had a characteristic phenotype of Andersen-Tawil syndrome, with short stature, hypertelorism, shield-shaped thorax, clinodactyly of the V finders on the hands and feet, and mild scoliosis. A potassium-enriched diet and 300 mg of potassium were recommended, and this was sufficient to preserve potassium levels at the higher physiological level (4.4–4.8 mmol/L). No muscular weakness episodes were reported later on. The treatment choice was β-blockers (metoprolol) and ICD. Several appropriate shocks were registered during seven years of follow-up. She had a normal pregnancy with continuous β-blocker therapy during the follow-up period. The patient canceled β-blocker therapy independently at the second trimester of the pregnancy, but frequent symptomatic PVC appeared, and β-blockers were restarted.

Genetic screening revealed heterozygous variant c.635G>A (p.Arg218Gln) in the *KCNJ2* gene, a known cause of Andersen-Tawil syndrome, and additional rare variant p.Thr983Ile in the *KCNH2* gene. Our group published this observation as a brief clinical case [14], and we assessed the p.Thr983Ile variant as likely pathogenic (Class IV) based on ACMG2015 criteria [13]. We then performed re-assessment of this variant with new recommendations refined for primary arrhythmias (2021) [15]. According to these latest recommendations [15], variant p.Thr983Ile does not meet PM2 criterion refined for LQTS (a rarity in population, 0.0001344 in gnomAD exomes, FAF > 5.5 × 10^−5^), and we did not find a clear track of five unrelated probands (to apply PM1 criterion). Strictly speaking, we have to re-assess this variant as VUS (a variant of unknown significance, Class III).

Unfortunately, the family history did not allow us to perform genetic testing on parents and determine the origin of both variants. However, genetic testing of offspring was performed and revealed the absence of both variants in a newborn.

Proband LQTS_139, female, 15 y.o., competitive athlete (swimming), without a history of syncope, no complaints, was diagnosed with QTc prolongation during routine ECG screening. Resting QTc was 449–480 ms, and provocative testing by standing was performed [16]. Standing ECG showed marked prolongation of the QTc from 449 ms to 564 ms (+115 ms), suggesting primary LQTS [16]. Potassium level was 4.7 mmol/L.

Genetic variant c.277G>A (p.Val93Ile), previously described as a cause of atrial fibrillation [17], was found in proband and her father (Figure 2). This variant was initially classified as likely pathogenic (Class IV) [17]. Additional genetic testing (whole-exome sequencing) was performed for the LQTS_139 proband to search for the possible genetic cause of QTc prolongation in other genes. A rare heterozygous variant p.Arg132Trp in the *SCN3B* gene of unknown significance (Class III) was found in the proband and her father.

Father (41 y.o), no complaints, was invited for the clinical examination, and borderline QTc prolongation (457 ms on resting ECG) was revealed on resting ECG. However, standing ECG had shown only a slight increase in QTc (+10 ms on standing ECG, 467 ms) (Figure 3A,B). No history of atrial fibrillation, facial or skeletal features of Andersen-Tawil syndrome was found in his brother and parents (all refused genetic testing).

We performed an independent functional analysis for the p.Val93Ile variant to confirm the initial study [18] about the gain-of-function effect of this variant on the Kir2.1 channel.

The IK1 current was induced by changing the membrane potential by a linear protocol from +60 to −120 mV: an abrupt shift from the maintained potential −80 mV to +60 mV, followed by a linear shift to −120 mV. In the course of voltage shift, a current with a characteristic IK1 I/V dependence was recorded [1].

At potentials more negative than EK in both groups of cells, an incoming current with an almost linear dependence on potential was observed. Conversely, at potentials more positive than EK, the outward current decreases as the potential approaches 0 mV and is practically absent at positive values of the potential. Next, the normalization of IK1 amplitude was performed by the value of the insignificant incoming component at −120 mV [1].

The outward component of the normalized current in cells expressing the mutant gene was significantly greater than in the control group of cells (*n* = 10, *p* < 0.05), at −40 and −50 mV (Figure 4A,B). This difference clearly suggests that the p.Val93Ile variant in the *KCNJ2* gene leads to the amplification of the output current component of IK1, and, thus, p.Val93Ile genetic variant is a gain-of-function mutation.

Forty-one alleles with this variant, including one homozygous case, were reported in gnomAD database, with total MAF 0.0001449 and predominant age of carriers from 40 to 80 years. This prevalence only slightly exceeds the threshold (0.01%) recommended in ACMG (2015) for dominant variants [13]. Predictive in-silico analysis suggests a benign effect of this variant from MutationAssessor, MutationTaster, PolyPhen2, DANN, EIGEN, and LIST-S2 (BP4 criterion). This amino acid position is moderately conserved through evolution, and p.Ile93 is presented in different species (Figure 5). Based on new refined recommendations, this variant unequivocally meets only two criteria: PS3 (reproducible data in vitro) and BP4 (most of in silico tools predict benign effect). Taking into account all data, we re-assessed p.Val93Ile as a variant of unknown significance (VUS, class III).

## 4. Discussion

The genetic causes of Andersen-Tawil syndrome are not entirely understood. Mutations in the *KCNJ2* gene account for only 50% of patients with a classic triad of symptoms: ventricular tachycardia and QTc prolongation, characteristic craniofacial abnormalities, and periodic skeletal muscular weakness [19]. The total prevalence of the *KCNJ2* gene mutations in LQTS patients seems to be low all over the world: 0.74% mutation carriers was reported in a large cohort of LQTS patients in the US [20], which was not very different from the Australian LQTS cohort (0.93%) [21] and low prevalence (1%) in this study. We assume that the real detectability of LQT7 in Russia might be even less. Taking into account the population count in Russia of 146 million (https://countrymeters.info/ru/Russian_Federation (accessed on 31 January 2022)), and prevalence of LQTS of 1:2500–1:2000 [22], we would expect 580–730 patients with LQT7 all over the country. However, we were only able to find one additional paper (Maltese et al., 2016) reporting a Russian patient with *KCNJ2* rare genetic variant (p.Arg82Gln), besides this study [23].

Not all patients with pathogenic/likely pathogenic variants in the *KCNJ2* gene express all clinical features of ATS [24]. It is assumed that clinical phenotype correlates with ion permeability defects. The loss-of-function (LoF) mutations of the *KCNJ2* gene are known to be causative for LQTS with variable skeletal and muscular phenotype and catecholaminergic polymorphic ventricular tachycardia [24], and gain-of-function (GoF) mutations can cause atrial fibrillation and Short QT syndrome [1,9,24]. No extra-cardiac effects of Kir2.1 variants for gain-of-function mutations have been reported. Our data agree with this observation: only probands with reported LoF and dominant-negative mutations (p.Arg67Trp and p.Arg218Gln) have additional skeletal abnormalities. Variant p.Arg67Trp affects relatively short N-terminus. Two variants in position 67 (p.Arg67Trp and p.Arg67Gln) were described, and clinical phenotype included extra-cardiac features [25]. Variant p.Arg218Gln localizes in a longer C-terminus domain, which forms the characteristic cytoplasmic extended pore domain [5,25]. Both variants localized in the intracellular domains of the Kir2.1 channel where other ATS mutations were described [5,7,25].

It seems that dysmorphic features in ATS are more reproducible for particular mutations (up to 88% in mutation carriers), whereas neurological and cardiac symptoms are more variable [26]. The possible explanation could be that potassium balance and cardiac repolarization is an ever-changing equilibrium in a very complex genomic context and dynamic environment, but the skeletal features are the result of the completed process.

It is known that Kir2.1 channels are important for craniofacial development [6], but the exact role of the current or molecule by itself is not entirely understood. The Kir2.1 channel was found to be a specific target in manifestation in fetal alcohol spectrum disorders (FASD), sharing some similar craniofacial appearance with ATS [27]. A study presents experimental evidence on Xenopus embryos that skeletal abnormalities associated with ATS are initiated during the early embryonic stage of development and are caused by the effect of potassium channel malfunction on the spatial distribution of Vmem of cells in the anterior ectoderm [6]. However, this study does not explain why some LoF mutations affect Vmem, and others do not, and why no visible dysmorphic consequences have been reported for GoF Kir2.1 alterations.

The influence of the complex genetic context, such as the role of additional rare genetic variants, requires more detailed fundamental research. For example, it was shown that rare variants in other genes encoding cardiac ion channels could modulate cardiac manifestation in *KCNJ2* mutation carriers [26].

Here, we present data of proband LQTS_33 carrying well-known genetic variant p.Arg218Gln in the *KCNJ2* gene. However, attention is drawn to cardiac manifestations that are unusually severe for the syndrome—a pronounced lengthening of the QTc interval, syncopal episodes, and at least one episode of muscular weakness, as well as a decrease in the level of potassium in blood plasma. Despite β-blocker therapy, the patient experienced several appropriate ICD shocks. It was shown that even common genetic polymorphism in the *KCNH2* gene may contribute significantly to the clinical appearance of cardiac arrhythmia in patients with *KCNJ2* mutations, and may serve as a predictor of symptomatic arrhythmias [26]. Highly symptomatic cardiac arrhythmia in LQTS_33 proband is in accordance with the published observation [26]. An additional genetic variant p.Thr983Ile in the *KCNH2* gene might represent a low-penetrant but important risk factor aggravating clinical phenotype. Interestingly, Maltese (2016) reported another Russian patient with clinically evident LQTS, QTc up to 530 ms, syncope, and ICD implanted, who was also the double-heterozygous carrier of the p.Arg82Gln variant in the *KCNJ2* gene and additional low-penetrant allele p.Arg176Trp in the *KCNH2* gene [23].

The most intriguing observation of this study is the identification of the GoF mutation p.Val93Ile, placed in the M1 transmembrane region of Kir2.1, in proband LQTS_139, a patient without any phenotypic signs of ATS and mild but evident QTc prolongation. There are several pathogenic variants affecting the M1 region (aminoacids 82–106), but they all increase QTc interval duration [7,28]. Electrophysiological study was consistent with previously published data [1] and confirmed the GoF mechanism. We also found a combination of p.Val93Ile in the *KCNJ2* gene and p.Arg132Trp in the *SCN3B* gene. However, we did not observe atrial fibrillation in either the proband or other family members. The genetic variant p.Arg132Trp in the *SCN3B* was registered in ClinVar (RCV000800514.1) as VUS (Class III) and did not clarify the clinical picture. Mutations in *SCN3B* were previously reported earnestly in patients with Brugada ECG pattern, IVF, and a case of SIDS [29]. We found the only mention of the p.Leu10Pro variant in the *SCN3B* gene in the context of LQTS in proband with postpartum cardiomyopathy and proband’s brother with normal QTc (402 ms), but no evidence of causality/pathogenicity was provided [30]. However, there is insufficient data to consider the p.Arg132Trp to be causative or modulating for the patient’s phenotype.

When assessing the pathogenicity of new variants, functional studies in model cells are of great importance. The presence of a confirmed significant effect on protein is a strong criterion (PS3) in favor of the pathogenicity of new unexplored variants, which, in combination with the criterion of rarity (PM2), almost immediately shifts the pathogenicity status to likely pathogenic. However, whether the cellular phenotype is enough to draw conclusions about disease phenotype is questionable. The functional analysis results of the p.Val93Ile variant are reproducible, making it possible to conclude a stable GoF mechanism, but three generations of pedigree LQTS_139 points that presence of this variant does not necessarily lead to atrial fibrillation.

Since first publication of p.Val93Ile in 2005, several genetic databases such gnomAD (https://gnomad.broadinstitute.org (accessed on 31 January 2022)) became available, which facilitates and significantly improves the process of variant assessment.

In proband LQTS_139, we observed discrepancy between clinical and experimental data. It is necessary for such discrepancies to be discussed; additionally, it underlines the importance of periodic reassessment of the clinical significance of even known variants. Several studies addressed the variant reassessment in cardiomyopathies [31,32], therefore upcoming re-evaluation of genetic variants in arrhythmias might be expected.

Variant p.Thr983Ile in the *KCNH2* was also re-assessed in light of current guidelines [15].

In our opinion, this variant should be classified as a pro-arrhythmogenic risk factor, similarly to what was proposed for certain other variants (p.R176T-*KCNH2*, p.D85N-*KCNE1*, etc.) [33]. This category is missing in the current recommendations, but we do agree with authors [33] that it could be very useful in clinical reality, along with a specific, agreed-upon protocol of genetic counseling (also missing at that time).

## 5. Conclusions

LQT7 is an exclusively rare form of LQTS in Russia, and accounts for about 1% of our LQTS cohort. We suggest that this sub-type of LQTS, with or without ATS phenotype, is largely underdiagnosed in Russia, and that both cardiologists and neurologists need more information about this syndrome.

Genetic variations in the genes encoding other cardiac ion channels may contribute significantly into the phenotype of patients with mutations in the *KCNJ2* genes, and may facilitate clinical appearance of arrhythmias.

Mutation p.Val93Ile variant leads to a gain-of-function effect in the different cell lines, but, surprisingly, the clinical appearance is not so consistent. This is a good example of the importance of periodic re-assessment of the pathogenicity of rare variants. Understanding of the genetic and cellular mechanisms of primary channelopathies is constantly growing. However, the knowledge gap in our understanding of the genotype-phenotype relationships still exists. We assume that multiple levels of evidence and reproducibility of data are crucial to our comprehension of the disease mechanisms and to gaining new insights into their clinical applications.

## Figures and Tables

**Figure 1 genes-13-00559-f001:**
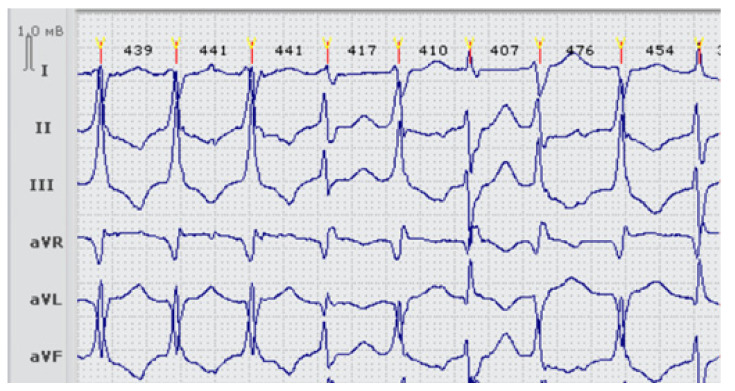
Fragment ECG of the proband LQTS33 (carrier of two heterozygous variants p.Arg218Gln in the *KCNJ2* gene and p.Thr983Ile in the *KCNH2* gene) at the first admission, before treatment. Frequent premature ventricular beats (>11,000 per day).

**Figure 2 genes-13-00559-f002:**
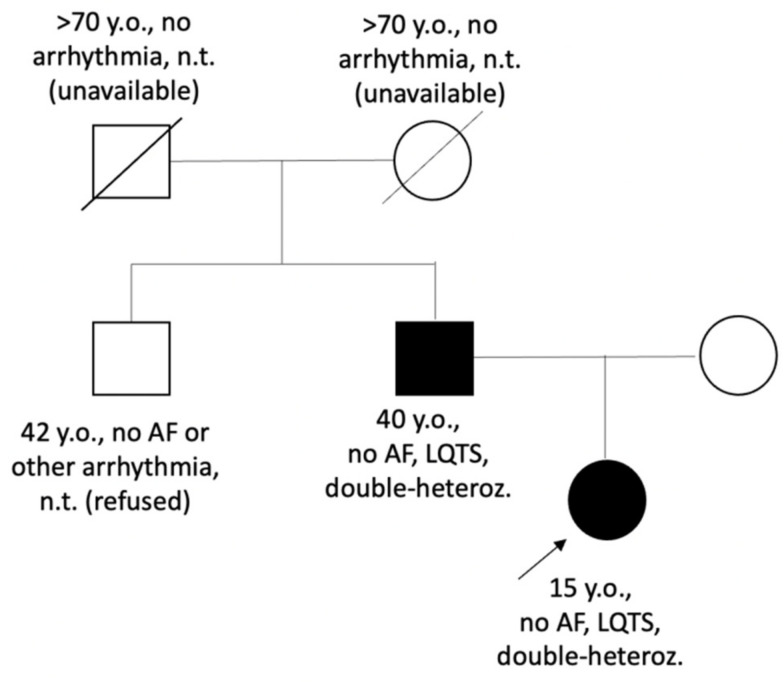
Pedigree of the LQTS_139 family. Circle symbols represent females, square symbols represent males. Closed symbols represent family members with QTc prolongation; open symbols—family members without known cardiac phenotype/diagnosis, proband is marked by arrow. AF—atrial fibrillation; double-heteroz.—carrier of two heterozygous variants p.Val93Ile in the *KCNJ2* gene and p.Arg132Thr in the *SCN3B* gene; LQTS—Long QT syndrome, n.t.—not tested.

**Figure 3 genes-13-00559-f003:**
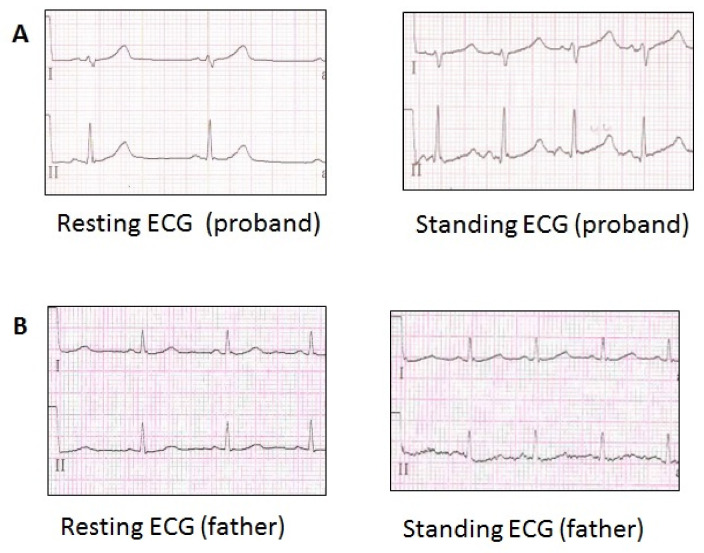
ECG of the family LQTS_139 members, carriers of the p.Val93Ile variant in the *KCNJ2* gene. (**A**) ECG fragment of the proband (Resting HR 59 bpm, QTc 449 ms), (Standing HR 103 bpm, QTc 568 ms). (**B**) ECG fragment of the proband’s father (Resting HR 78 bpm, QTc 458 ms), (Standing HR 97 bpm, QTc 467 ms).

**Figure 4 genes-13-00559-f004:**
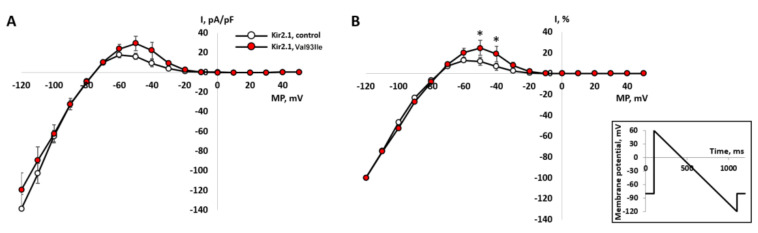
Background inward rectifier potassium current IK1 recorded in CHO-K1 cells transfected with the wild-type gene of Kir2.1 channel (n = 10) and the mutant gene (n = 9). The current was elicited by hyperpolarizing ramp protocol (see inset) from the holding potential of -80 mV. (**A**)—I-V curves of IK1 in absolute values of current density. (**B**)—comparison of normalized I-V curves in % of inward current maximum measured at −120 mV. The current values for each cell were normalized independently and averaged after normalization. *—significant difference mutant vs. control, *p* < 0.05, two-way ANOVA with Tukey post hoc test.

**Figure 5 genes-13-00559-f005:**
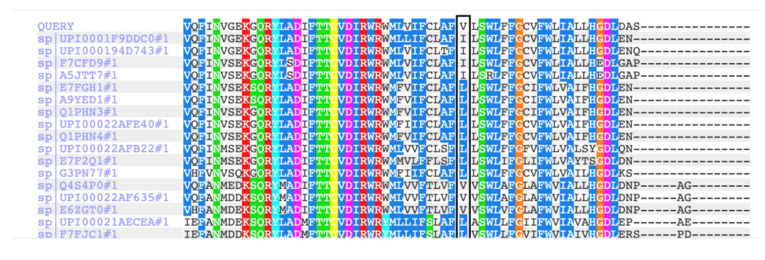
Multiple alignment of the Kir2.1 protein sequencing in different species (by PolyPhen2.0). Position 93 in the Kir2.1 is moderately conserved. Variant Val93 presented in 22 species (79%), variant Ile93—in 6 species (21%).

**Table 1 genes-13-00559-t001:** Clinical and genetic characteristics of the probands with rare heterozygous variants in the *KCNJ2* gene.

Family (Ptnt)	Age *, Years	Variant/ (Class of Pathogenicity)	Origin	Kir2.1 Current Effect	QTc Resting	QTc Standing	Syncope	Andersen-Tawil Phenotype	Arrhythmia	Therapy
LQTS_96	29	p.Arg67Trp (V)	De novo	Decreased [9]	494–500	540	Yes	+	VT, PVB, bigeminy	BB, ICD
LQTS_139	15	p.Val93Ile (III), p.Arg132Trp in *SCN3B* (III)	Inherited	Increased [11,13]	449–480	568	No	no	no	BB recommended (refused)
LQTS_33	23	p.Arg218Gln (V), p.Thr983Ile in *KCNH2* (III)	DNA samples from parents unavailable	Decreased [14]	600–620	650	Yes	+	PVB, paroxysmal VT, VF, appropriate shocks	BB, ICD

BB—β-blockers, ICD—implantable cardioverter-defibrillator, PVB—premature ventricular beats, VF—ventricular fibrillation, VT—ventricular tachycardia. *—The age is marked at the time of the first genetic consultation and DNA testing.

## Data Availability

The clinical data presented in this study are available on request from the corresponding author. Data collected for this study are available through the publicly database ClinVar (Accession numbers: SCV002038508, SCV002038509, SCV002039183, SCV002041901).

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
