# Peer review of "Variable Clinical Appearance of the Kir2.1 Rare Variants in Russian Patients with Long QT Syndrome"

_genes, 2022, doi:10.3390/genes13040559_

Round 1
Reviewer 1 Report
The manuscript by Elena Zaklyazminskaya et al. reported a study of a Kir2.1 genetic screening in Russian patients diagnosed with LQTS. The authors sequenced the over 200 patients’ DNA and found 3 independent variants in KCNJ2 which were previously reported and assessed based on ACMG2015 criteria. This number agrees with the low prevalence of ATS in other parts of the world. Among the 3 cases, 2 of them have ATS (R67W&R218Q), and one only has long QTc (V93I). The author further performed electrophysiology to demonstrate V93I is a gain-of-function mutation. The author then reassessed the mutation as VUS with clinical data. The reassessment needs a little bit more evidence.
The current study is on a topic of relevance and general interest to the readers of the journal, especially for the Human Genomics and Genetic Diseases section. The author presented interesting clinical and genetic data. I found the paper to be overall well written.
Major comments:
- In the introduction part, the author did not mention the structure/morphology of Kir2.1 channel. If the authors provide the structural information in the introduction, when the authors mention the three mutations found in the genetic screening later, it would be easy to add that V93 is in the transmembrane domain while R67 and R218 are in the cytoplasmic domain. Also, it might be beneficial to include a mapping of known KCNJ2 disease mutations on its structure in the introduction. The paper ‘Disease Associated Mutations in KIR Proteins Linked to Aberrant Inward Rectifier Channel Trafficking’ on Biomolecules in 2019 or ‘Phenotype Variability in Patients Carrying KCNJ2 Mutations’ on Circulation: Cardiovascular Genetics in 2012 have figures like that.
- In the introduction part, the author did not mention KCNJ2 inheritance traits.
- In the Materials and Methods part, the author mentioned the study was in accordance with the Helsinki declaration but did not mention whether the study was approved by the ethics committee.
- In the Materials and Methods part, the author did not mention where genomic DNA was extracted and how the DNA purification was carried out.
- In the Materials and Methods part, the author did not describe the whole cohort of probands and did not mention that first-degree and second-degree relatives of some patients were tested.
- In the Materials and Methods part, the author mentioned that they used GRCh37 build instead of the latest GRCh38. Is there a reason for this?
- In the Materials and Methods part, the author mentioned that 10 other genes were target sequenced as well. It’s important to clarify if there were additional genetic findings in probands. Did they exclude the unexpected presence of compound mutations related to primary electric diseases?
- Qualification of the variants should be provided with more details. How the V93I was reassessed as VUS? What analysis was used? Were the 6 species presenting I93 evolutionarily further apart from human than the other 22 species? Also, the gain-of-function of KCNJ2 can cause Short QT syndrome and AF. Can the low prevalence come from the fact that the cohort being studied has LQTS instead?
- In the Materials and Methods part, the author mentioned that they performed clinical investigations on all probands. Were there other ATS patients which did not have KCNJ2 mutation detected in this study?
- In the discussion part, the author jumped from LQTS mutation to ATS prevalence. However, not all KCNJ2 mutations contribute to ATS and not all ATS patients have KCNJ2 mutations. Please clarify how the number 850-1800 was calculated.
- The title of this paper is for all LQTS and the author did genetics screening on all LQTS patients. It might be better not to focus on ATS part in the conclusion (the same reason as 9).
Minor comments:
- There are abbreviations that need to be explained in the text:
line27: ATS is Andersen-Tawil syndrome
line43: EK is potassium equilibrium potential
- There are some spell/grammar errors:
line29:’COS-K1’ should be ‘CHO-K1’
line46: ‘mutlisystem’ should be ‘multisystem’
line48&49: should be ‘maintains’ and ‘plays’
- line47: The cytogenetic location of KCNJ2 on ref2. is 17q23.1-24.2. Also, this mapping needs to have a reference.
- line 51 is a duplicate of line 48.
- The introduction part needs more references. For example, line 53 needs references for pro-arrhythmic disorder cases from KCNJ2 mutations.
- line 79: MGISEQ-2000 and DNBSEQ-G400 are two models from MGI for sequencing. Please clarify which one was used.
- Patient 139’s father is 41 years old in Table 1 and 40 years old in Figure 2. Also, the patient’s father has SCN3B mutation as the text indicated but the table did not mention it. It might be better to leave out all the father’s information in table 1 and only include probands for this table.
- Figure2 did not mention the gender of patients. Is circle female?
- The QTc resting values in Table1 are all different from the range mentioned in the text.
- line330: only 2 patients have ATS. How was 1.4% calculated?
Reviewer 2 Report
The raport seems to be interesting but the conclusions are not clinically attractive. The authors need to reconsider their conclusions.
Reviewer 3 Report
I would like to start by saying its an interesting read, It has great potential to be a high impactful article, nonetheless I feel that there is a need to revise the flow of the article it self..
p.Arg67Trp, p.Val93Ile, and p.Arg218Gln, have been previously described, please revise the following literature to add and or edit elements to your manuscript, which will help enrich it particularly in the intro and discussion sections
https://link.springer.com/article/10.3103/S0096392521030056
https://doi.org/10.1016/S0092-8674(01)00342-7
https://www.nature.com/articles/ejhg2013139.pdf?origin=ppub
https://www.ncbi.nlm.nih.gov/books/NBK1264/
https://www.sciencedirect.com/science/article/pii/S0914508717300655
https://bmcneurol.biomedcentral.com/articles/10.1186/s12883-019-1322-6
Here is my issue, you mention the 3 cases of rare mutations but from the reader stand point you have no reason since they are not mentioned in the intro as of potential importance, particularly because other mutations may also be of importance for a specific reader. This is an issue as currently stated in the beginin of the results were you say We found three LQTS probands carrying rare heterozygous genetic variants 109 p. Arg67Trp, p. Val93Ile, and p. Arg218Gln in the KCNJ2 gene… so what about the other cases.. no mutations or not important, and if the idea is to connect to Andersen-Tawil (which makes sense) describe the connection of the mutations to the syndrome in the intro o discussion.
In lines 178 you make a mention about the SCN3B which could be related to Brugada which give issues in Nav, would be nice to have more mention of similarities (elsewhere in the manuscript).
Now on the other side I find it interesting that you chose Val93Ile mutation only, while this is smart to focus on a particular mutation, I refer back from the intro no reason for it and one would even say, why not do all 3 and present them in the same paper. Hence, revise to guide the reader in a smoother sense
The genetic study for all probands included target genes panel sequencing for 11 genes (KCNQ1, KCNH2, SCN5A, KCNE1, KCNE2, KCNJ2, SNTA1, SCN1B-4B genes)…. This is mention but where are the results or the discussion of the same, no mention is done, hence where they done? Was there any analysis?
Round 2
Reviewer 3 Report
I thank the authors for the well versed improvements